# Attenuation Correction of Long Axial Field-of-View Positron Emission Tomography Using Synthetic Computed Tomography Derived from the Emission Data: Application to Low-Count Studies and Multiple Tracers

**DOI:** 10.3390/diagnostics13243661

**Published:** 2023-12-14

**Authors:** Maria Elkjær Montgomery, Flemming Littrup Andersen, Sabrina Honoré d’Este, Nanna Overbeck, Per Karkov Cramon, Ian Law, Barbara Malene Fischer, Claes Nøhr Ladefoged

**Affiliations:** 1Department of Clinical Physiology and Nuclear Medicine, Rigshospitalet, Copenhagen University Hospital, 2100 København, Denmark; maria.elkjaer.montgomery@regionh.dk (M.E.M.); nanna.overbeck.petersen.01@regionh.dk (N.O.); per.cramon@regionh.dk (P.K.C.); ian.law@regionh.dk (I.L.); barbara.malene.fischer@regionh.dk (B.M.F.); claes.noehr.ladefoged@regionh.dk (C.N.L.); 2Department of Clinical Medicine, Copenhagen University, 2200 København, Denmark; 3Department of Applied Mathematics and Computer Science, Technical University of Denmark, 2800 Lyngby, Denmark

**Keywords:** LAFOV, PET/CT, attenuation correction, deep learning, motion correction

## Abstract

Recent advancements in PET/CT, including the emergence of long axial field-of-view (LAFOV) PET/CT scanners, have increased PET sensitivity substantially. Consequently, there has been a significant reduction in the required tracer activity, shifting the primary source of patient radiation dose exposure to the attenuation correction (AC) CT scan during PET imaging. This study proposes a parameter-transferred conditional generative adversarial network (PT-cGAN) architecture to generate synthetic CT (sCT) images from non-attenuation corrected (NAC) PET images, with separate networks for [^18^F]FDG and [^15^O]H_2_O tracers. The study includes a total of 1018 subjects (*n* = 972 [^18^F]FDG, *n* = 46 [^15^O]H_2_O). Testing was performed on the LAFOV scanner for both datasets. Qualitative analysis found no differences in image quality in 30 out of 36 cases in FDG patients, with minor insignificant differences in the remaining 6 cases. Reduced artifacts due to motion between NAC PET and CT were found. For the selected organs, a mean average error of 0.45% was found for the FDG cohort, and that of 3.12% was found for the H_2_O cohort. Simulated low-count images were included in testing, which demonstrated good performance down to 45 s scans. These findings show that the AC of total-body PET is feasible across tracers and in low-count studies and might reduce the artifacts due to motion and metal implants.

## 1. Introduction

Recent developments in PET/CT detectors have introduced SiPM-based detectors, pushing the limit for time of flight (TOF) towards 200 ps [1], thereby increasing the signal-to-noise ratio and increasing the image quality and lesion detection [2,3]. With the emergence of long axial field-of-view (LAFOV) PET/CT scanners such as the Siemens Biograph Vision Quadra, Siemens Healthineers [4], and the uEXPLORER, United Imaging [5], the combination of detector coverage of the whole body and the high TOF resolution has increased PET sensitivity markedly allowing for a drastic reduction in the injected tracer dose by a factor of ten or more [6]. This has potential implications for easing the path to scanning subjects where a reduced radiation dose is desired: pediatric patients, pregnant women, patients in repeated control schemes such as malignant melanomas, or research projects with healthy controls. Several corrections are required for the accurate quantification of PET images, including random coincidence events, detector dead time, crystal efficiency normalization, and correction for Compton scatter events and attenuation. Scatter and attenuation correction relies on a knowledge of object geometry and the electron density of tissue commonly acquired by an attenuation correction CT scan (AC CT) as part of the PET/CT examination [7]. Moving towards PET scanning with a very low patient radiation dose (<0.5 mSv) [8,9], the associated AC CT scan will become the primary source of ionizing radiation for the patient as even a low dose CT routinely used for attenuation correction and localization contributes by more than a factor of 10 (5–7 mSv) [10].

Artificial intelligence has shown great potential for synthesizing high-quality CT data (sCT) from Magnetic Resonance Imaging (MRI) using deep learning based on convolutional neural networks (CNNs) or generative adversarial networks (GANs) [11,12,13,14,15]. While initial work targeted synthesizing sCT for the brain, recent works have addressed the more complex tissue composition and geometry of whole-body imaging [16,17]. For total body PET imaging using a LAFOV PET/CT scanner, MRI data are typically not available or would pose a challenge due to differences in patient positioning during scanning. Therefore, a solution to attenuation correction where generated attenuation maps and emission data are positioned alike is desired [18]. One approach is the direct estimation of attenuation and scatter-corrected PET images from the non-attenuated emission data (NAC PET) using CNNs [19,20] or cycle-consistent GAN (CycleGAN) networks [21], thereby bypassing the normal iterative PET reconstruction pipeline. This approach shows impressive results, but it might be difficult to assess whether the transformation from NAC PET to the final PET images is indeed correct on an individual basis since GAN networks can be prone to mode collapse, causing them to generate the same output from different inputs [22]. Another approach would be to synthesize AC CT images from the NAC PET. This approach increases explainability as the PET reconstruction is still handled by the vendor-implemented reconstruction method of choice, and artifacts in the sCT can be assessed before PET reconstruction, which allows for a quality assurance check by scanner staff and physicians that fits well into a routine clinical pipeline. A number of studies have used this approach [23], showing promising results towards accurate attenuation correction without CT. These previous studies typically have datasets in the range of 25–220 that might be adequate for training a CNN, but increased sample size should increase robustness towards variation in body shapes, artifacts, and rare anatomy, as new patients are more likely to be represented in the training cohort [15]. Challenges are reported especially in the lung region where little NAC PET signal is present [21,24]. Furthermore, deriving sCT from NAC PET has the advantage of addressing movement artifacts due to differences in homologous tissue during PET and CT acquisition. This is often observed as the so-called banana artifact around the diaphragm related to differences in the breathing cycle, but also related to bowel motions as well as gross body motion. Finally, artifacts due to streaking and beam hardening artifacts from metallic implants that might influence PET quantification [25] could be addressed by deriving the attenuation map from the PET emission data.

We propose a deep learning-driven synthetic CT generation procedure where sCT images are produced directly from the non-attenuation corrected PET images, thereby eliminating the need for a separate CT scan on LAFOV PET/CT scanners. We furthermore validate the robustness of the derived model for variation in PET radiotracer and count rate to address the need for differences in clinical protocols and pave the way for ultra-low-dose (<0.5 mSv) PET scanning.

## 2. Materials and Methods

### 2.1. Patient Cohort

This retrospective study consists of five cohorts with 1018 subjects in total. The proposed algorithm was developed using 858 consecutively included subjects injected with [^18^F]FDG and scanned on one of four separate Siemens Vision 600 PET/CT scanners. The algorithm was evaluated using a separate cohort of [^18^F]FDG-PET/CT data acquired after the training data acquisition period on either the same scanner (*n* = 78) or on a LAFOV Siemens Vision Quadra PET/CT (*n* = 36). We also included 46 subjects injected with [^15^O]H_2_O and scanned on the LAFOV PET/CT. We used 34 of these subjects to transfer-learn the original algorithm to the new radiotracer and the remaining 12 subjects to evaluate the algorithm. All patient-specific data were acquired at Rigshospitalet (Copenhagen, Denmark) and were handled in compliance with the Danish Data Protection Agency Act no. 502, including full anonymization. The cohorts are listed in Table 1.

### 2.2. Data Acquisition

The data acquisition differed between and within the cohorts but always consisted of an NAC PET and CT pair. The [^18^F]FDG training cohort was made up of *n* = 858 subjects scanned with (*n* = 620) or without (*n* = 238) IV contrast. All subjects in the [^18^F]FDG test cohorts were scanned with CT without IV contrast. Data were acquired using CarekV at 120 ref. kVp, and with dose modeling using careDose at 170 ref. mAs. The CT images were reconstructed at 512 × 512 matrices. [^18^F]FDG images were acquired according to the European guidelines, with [^18^F]FDG administered at 3 MBq/kg body weight 60 min prior to scanning. Subjects were scanned with either arms up or arms down. NAC PET images were reconstructed without attenuation correction using 3D ordinary Poisson OSEM (3D-OP-OSEM). No point spread modeling (PSF) was applied, and post-filtering was set at 4 mm. All PET images had a voxel size of 1.65 × 1.65 × 2 mm^3^ (440 × 440 matrices). The training cohorts and Vision 600 test set were reconstructed at the PET/CT scanner. The test LAFOV cohorts were reconstructed offline using e7tools (Siemens Healthineers, Knoxville). In addition, we simulated the effect of reduced scanning time by reconstructing the LAFOV [^18^F]FDG test set in frames of 30 s, 45 s, 90 s, 180 s, and 300 s. 

The [^15^O]H_2_O cohorts consisted of patients examined with two different protocols: a clinical cerebral blood flow (CBF) dynamic imaging protocol for patients with steno-occlusive disease and an ongoing protocol evaluating multi-organ perfusion in patients with thyroid disease (Protocol no. H-21034679). CT scans were acquired with a slice thickness of 3 mm (initial 7 subjects at 1.5 mm). Furthermore, eight of the training subjects were acquired with a low-dose CT (ref. mAs 30) before switching to an ultra-low-dose CT protocol (ref. mAs 7). Data were acquired from the start of [^15^O]H_2_O injection for a duration of 4–12 min. In the clinical CBF protocol, scans were repeated both prior to and after the injection of acetazolamide (Diamox, Amdipharm, Helsingborg, Sweden), i.e., 2–4 datasets were available for each subject. For the thyroid perfusion examination, two datasets were available for each subject. Thus, the total number of datasets for training and testing was *n* = 106 and *n* = 41, respectively. We used a static 3 min reconstruction no-PSF, from 1 to 4 min post-injection to exclude the initial vascular phase of [^15^O]H_2_O NAC PET. The reconstructions were performed offline using e7tools.

### 2.3. Pre-Processing

Pre-processing steps were identical for [^18^F]FDG and [^15^O]H_2_O images: CT images were resampled to have pixel dimensions 2 × 2 × 2 mm^3^. To unify the input to the model and reduce the computational load, the CT images were cropped before resampling in order to exclude air outside the patient using a threshold of −400 HU for air. The NAC PET images were resampled to the cropped 2 × 2 × 2 mm^3^ CT images. The CT images were then normalized by Equation (1), and the NAC PET images were normalized by Equation (2) where p_0.5%_ and p_99.5%_ are the 0.5% and 99.5% percentile of the image.
(1)$${CT}_{norm}=\frac{CT+1024}{2000}$$
(2)$${PET}_{norm}=\frac{PET-p_{0.5\%}}{p_{99.5\%}-p_{0.5\%}}$$

### 2.4. Synthetic CT Generation

We proposed a 3D parameter transferred conditional GAN (PT-cGAN) network architecture. We trained a PT-cGAN model for each tracer. In both models, the generator was first pre-trained using the large cohort of [^18^F]FDG training patients (with and without IV contrast, *n* = 858). The PT-cGAN model was then trained using tracer-specific cohorts. The FDG model included the same patients used during pretraining, but only the patients without IV contrast were included (*n* = 238). For the H_2_O model, this included the training H_2_O cohort (*n* = 34).

We utilized an all-convolution 3D U-net with filters (64, 128, 256, 512), which was trained with a mean absolute error (MAE) loss using the Adam optimizer and a learning rate of 2 × 10^−4^ for 300 epochs with batch size 8. In each epoch, a total of 12 random patches (128 × 128 × 32) were extracted for each patient. Random data augmentation (rotation, translation, scaling) was subsequently performed using TorchIO [26]. The final model was chosen as the model with the best validation loss.

The discriminator network in the PT-cGANs is a binary classifier consisting of 5 convolutional layers. The discriminator is conditioned on the NAC PET patch and was trained to determine if the given CT and NAC PET pair represented a real or synthetic CT. The output of the network indicates whether the input CT is real or fake (synthetic). To balance out the performance of the generator and discriminator, the discriminator was set to train separately for 50 epochs, after which the pre-trained generator and discriminator were trained in turn in an adversarial manner. The discriminator was trained using binary cross entropy as a loss function using the Adam optimizer. The learning rate was initially set to 2 × 10^−3^ and then dropped to 2 × 10^−4^ after 25 epochs. 

The generator was trained using a combination loss function consisting of MAE loss, $L_{MAE}$, for the entire image; the discriminator loss for synthetic input, $L_{disc}$; and a dice loss for bone with bone defined as values above 100 HU. This loss function, gen_loss_, is defined as Equation (3).
(3)$${Gen}_{loss}=L_{disc}+150\cdot L_{MAE}+1\cdot L_{dice\left(bone\right)}$$

The motivation behind this loss function was to optimize the generator on three parameters: the image quality of the generated images, the network’s ability to trick the discriminator, and the network’s ability to segment bones. The generator learning rate was set to 1 × 10^−4^. The cGAN framework was trained for 1500 epochs with a batch size of 8. The final model was then chosen using a combination of visual inspection and evaluation metrics based on dice value for tissue and bone, RMSE, and MAE. The PT-cGAN trained using the [^15^O]H_2_O cohort was trained using an increased generator learning rate (1 × 10^−3^ instead of 1 × 10^−4^) for a longer period (4000 epochs instead of 1500) to account for the change in tracer. Fewer patches were extracted per subject (4 instead of 12) to accommodate for the smaller cohort.

The synthesis steps for both models were identical. First, overlapping patches from the NAC PET image were sampled and given to the trained generators, which outputted the corresponding sCT patches. The sCT patches were then combined, where the average was taken whenever the patches overlapped. The complete sCT image was then de-normalized, padded, and resampled such that the combined sCT image had the size and dimensions of the original CT input. Finally, the bed from the original CT image was superimposed onto the sCT image. 

The proposed models were implemented in Python 3.9.7 using PyTorch Lightning (v 1.5.10) and trained on four NVIDIA V100 32GB GPUs. Synthesis was performed using a single NVIDIA Titan RTX 24 GB GPU, Nvidia, Santa Clara, CA, USA.

### 2.5. PET Reconstruction

Two PET reconstructions of each LAFOV test set patient were performed for evaluation of both tracers. The sCT was used for attenuation correction to generate sPET, and the standard CT was used to generate a standard PET image for reference. Reconstruction parameters were in line with our clinical routine settings using e7tools with 3D-OP-OSEM: 2 mm post filter and PSF for [^18^F]FDG and no-PSF, 4 mm post filter, static 1–3 min for [^15^O]H_2_O. 

In addition, the LAFOV [^18^F]FDG subjects were reconstructed for the full acquisition period (300 s) but using sCT from each of the time-reduced NAC PET images (sPET30, sPET45, sPET90, sPET150) to simulate shorter acquisitions.

Due to the retrospective study design, PET raw data were not available for reconstruction for the Vision 600 [^18^F]FDG test cohort.

### 2.6. Data Analysis

#### 2.6.1. Synthetic CT Analysis

For quantitative evaluation of the model predictions of accurate CT HU values, mean absolute error (MAE) and structural similarity index measure (SSIM) between sCT and CT were computed at the Vision 600 test set. This allows for a comparison of the sCT performance between the Vision 600 (used for training) and the LAFOV Vision Quadra used for test. We evaluated performance of each dataset using a *t*-test.

#### 2.6.2. Qualitative Analysis

The [^18^F]FDG sPET images were evaluated using the test LAFOV cohort. Qualitative evaluation was performed by visual inspection by an experienced nuclear medicine specialist who was presented with PET and sPET images blinded to the AC method. For each subject, the PET data were presented in Microsoft PowerPoint side by side in random order. All transaxial slices were available by scrolling, as were the maximum intensity projection (MIP) image with manual rotation. For each reconstruction, it was noted whether attenuation artifacts due to motion between PET emission and the attenuation map were present or if metal-induced artifacts could be seen in the PET data. Artifacts were rated using a Likert scale of 0 = none, 1 = minor (no clinical impact), 2 = medium, and 3 = major (potential clinical impact). Furthermore, it was noted if there were any differences in image quality and, if so, which image was superior. The image quality was further assessed on a 0–2 scale: (0 = same quality, 1 = insignificant difference without clinical impact, and 2 = significant difference with potential clinical impact). 

#### 2.6.3. Quantitative Analysis

Images were evaluated quantitatively by computing the relative mean difference between the sPET and the reference PET in different organs for both tracer cohorts. This was performed by deriving organ masks from CT using the segmentation prototype MIWBAS from Siemens Healthineers [27]. The liver, lungs, kidneys, heart, aorta, spleen, brain, and bones were evaluated.

#### 2.6.4. Robustness towards Reduced Count-Rate

Finally, we evaluated each sCT*x* and corresponding sPET*x*, where *x* refers to the reconstructed acquisition time (30 s, 45 s, 90 s, 150 s, 300 s), with a quantitative comparison to the full-count sCT/sPET and reference CT/PET pairs. Evaluation was performed for selected organs using the MIWBAS derived masked from the CT.

## 3. Results

### 3.1. Qualitative Evaluation

The visual inspection of [^18^F]FDG-PET vs. sPET showed no differences in image quality in 30/36 cases. Six cases showed minor insignificant differences, and no cases showed significant differences, see Table 2. Six patients showed motion artifacts near the liver. In all cases, these artifacts were removed or reduced in the sPET. Two subjects had artifacts deemed due to metal affecting the PET images, one due to a metal implant with no changes in clinical PET and one due to a hip implant where two small lesions were easily seen on sPET. 

A sample patient is shown in Figure 1, illustrating the banana artifact in the PET data (Figure 1b) due to the mismatch between the CT scan (Figure 1a) and emission data (Figure 1f). The synthetic CT (Figure 1c) was synthesized from emission data, and therefore, no mismatch was found, and the sPET appeared correct in the liver region (Figure 1d). This motion due to breathing affected the whole region around the diaphragm and resulted in markedly different PET values when using the anatomically matching synthetic CT for attenuation correction, visualized in Figure 1i. Figure 2 shows an example patient with severe CT streaking artifacts due to a double shoulder implant. We noted that the sCT did not have these artifacts. The same observation was seen in patients with hip or dental implants. The voxel-wise relative difference image (Figure 2c) visualizes the effect on the PET image.

### 3.2. Quantitative Evaluation

For the quantitative comparison between CT and sCT images for the [^18^F]FDG test cohorts, an MAE value of 21.28 ± 4.01 HU and an SSIM value of 0.95 ± 0.01 was obtained for the sCT images generated from the Vision 600 data. For the LAFOV cohort, MAE was 19.51 ± 3.62 HU and SSIM 0.96 ± 0.01. This difference was not significant at a 5% significant level.

The quantitative PET analysis of the [^18^F]FDG test cohort is shown in Figure 3. The mean relative error was below 5% for all organs except the brain at 8%. We found a mean relative error across organs at 1.18% including the brain and 0.45% excluding the brain. As the [^18^F]FDG training cohort was selected from the clinical routine production, most patients are scanned below the eyes only, and the scan did not include the full brain. This might affect the results from the brain region. 

The evaluation of the static PET water scans is summarized in Figure 3. The relative mean error was below 3.12% for all organs. Note that the brain was performing well on these LAFOV [^15^O]H_2_O scans, which might be due to the brain always being included in the training scans. Outliers were observed and expected as the motion between CT and PET can have a large impact at the patient level. An example patient is presented in Figure 4.

For the analysis of robustness towards the reduced count rate, Figure 5 shows the mean relative deviations for the chosen organs for each acquisition time. Excluding the brain that, as mentioned, was not systematically represented in the training data, we obtained good performance down to the 45 s scan, and even at 30 s performance, the mean relative error at the organ level is below 5% when further excluding bones. An example patient is shown in Figure 6 with varying frame sizes.

The performance of subjects with abnormal anatomy could be a concern. We refer to a sample subject in Appendix A, showing a WB [^18^F]FDG PET from the LAFOV test cohort from a patient with an amputated right leg. This is an extreme case; PET quality is well recovered, including the scoliosis and dislocation of internal organs.

## 4. Discussion

In this study, a deep learning approach for synthetizing CT images directly from PET emission data was implemented and evaluated with the purpose of eliminating the need for a separate attenuation scan. This will achieve two goals: (1) lower the radiation dose to patients where a clinical CT is not required and (2) address the artifacts related to the mismatch between PET and CT scans due to motion, most commonly respiratory or cardiac motion. The latter is a problem that was already pointed out with the introduction of the PET/CT scanner in 2001 [28]. Similarly, artifacts may arise from a mismatch of air in the stomach or bowel, which is displaced due to peristalsis between the PET and CT acquisitions [29]. By generating the attenuation map from the emission data, we ensured that PET and attenuation data were time-wise aligned and, at the same, in a homologous position. The proposed model was trained on 858 subjects injected with [^18^F]FDG and scanned on a Siemens Vision 600 PET/CT. The model was evaluated on 114 subjects from a separate cohort, which were either scanned on the same scanner (*n* = 78) (CT vs. sCT evaluation only) or on an LAFOV PET/CT (*n* = 36) (full PET vs. sPET evaluation). To evaluate robustness towards lower count rates, we tested the performance by stepwise shortening the reconstruction time. We found a mean relative error of 1.2% across all organs (liver, lung, kidney, heart, aorta, spleen, brain, and bone) included in the LAFOV dataset. When excluding the brain that was not systematically present in the training data, the error dropped to 0.5%. This is in line with results reported by other groups; Hu et al. [30] reported a mean error of 3.2% using a similar approach with a Wasserstein-based loss function, and Xue et al. [31], similar to our approach, trained a 2D GAN on short-axis FOV PET/CT data (*n* = 165) and obtained a normalized mean squared error of 0.5% across seven test subjects scanned on a LAFOV scanner. For the reduced count rate simulation, we found a slight decrease in performance when reducing frame size with optimal performance at 90 s corresponding to the noise level for the training data. Performance was found to be below 5% mean error (excluding the brain) down to 30 s with clinical [^18^F]FDG dosage (3 MBq/kg bw), indicating robustness towards variation in count rate due to variation in injected activity or scanning time.

To assess the model’s ability to perform on tracers other than [^18^F]FDG, we applied transfer learning to a cohort of subjects examined using ^15^O-water PET scanning. Forty-six subjects injected with [^15^O]H_2_O were scanned on a LAFOV PET/CT and split into a training (*n* = 34) and test (*n* = 12) set. Despite the small cohort, we found the mean error stayed below 5% except for bones (5.9%). In this model, the brain was included in all training scans, and this is apparent with a mean error of just 2.5%. Based on this, we would expect the [^18^F]FDG model to perform on par with the [^15^O]H_2_O-water model by including brains in the training.

Quantitative results for the sCT analysis showed low MAE for both the Vision 600 and the LAFOV [^18^F]FDG cohorts at 21.3 and 19.5 HU, respectively. The SSIM was 0.95 and 0.96, respectively, which is slightly below the result (0.98–0.99) reported by Hu et al. [30]. The CT image metrics showed no significant difference between the two cohorts, indicating that the Vision 600 model was suitable for inference of LAFOV NAC PET data.

The qualitative results demonstrated no significant difference between all 36 cases when rated by an experienced nuclear medicine physician. More specifically, 30/36 showed no qualitative difference at all. For the six cases with minor differences, half were rated with PET superior and half with sPET superior. In all cases with the banana artifact in the CT-based PET, the physician observed a reduction of the artifact in sPET. This qualitative assessment is a strong indication that the method is feasible and ready for clinical testing. Furthermore, case inspection (Figure 1) showed a large variation around the lower lung/diaphragm. Even though a ground truth is not available, the logical conclusion here is that the emission-derived sPET is quantitatively more accurate than the CT-based PET in an area of motion. Generating a synthetic CT for attenuation correction has the benefit of allowing for validation of AI-derived data before reconstruction and intervention if image artifacts are found. This is apparent in the case shown in Appendix A, where abnormal anatomy can be confirmed in the attenuation map and where an artifact in the form of an air pocket can be identified and addressed in the interpretation. We believe this leads to improved trust in an AI-based approach to CT-less PET scanning.

This study has some limitations. Firstly, due to the retrospective setup, no PET raw data were available for the large Vision 600 cohort. Furthermore, whole-body [^18^F]FDG PET was typically scanned below the brain in our institution; therefore, model performance at the brain level is challenged. For the [^15^O]H_2_O model, the sample size was only 12 subjects for testing. Even then, the results are promising and suggest that cross-tracer transfer learning from the large [^18^F]FDG cohort might be feasible. Furthermore, the reliability of the clinical output of [^15^O]H_2_O PET, namely the quantitative estimation of CBF and the flow response to acetazolamide [32], was not estimated per se.

## 5. Conclusions

In this manuscript, we have demonstrated that attenuation maps can be synthesized from [^18^F]FDG PET emission data, thereby eliminating the need for an attenuation CT scan. This has the benefit of reducing patient dose, even below 1 mSv, but also suppressing common artifacts in PET due to motion or metal implant artifacts in CT. Robustness towards variation in scanning time was shown, thereby accommodating variations in scanning protocols. Finally, transfer learning from the [^18^F]FDG model to a LAFOV [^15^O]H_2_O model was feasible even with a small sample size and good accuracy of the final PET reconstruction. This suggests that the proposed model can reduce the need for independent imaging of anatomy with application to CT-less scanning.

## Figures and Tables

**Figure 1 diagnostics-13-03661-f001:**
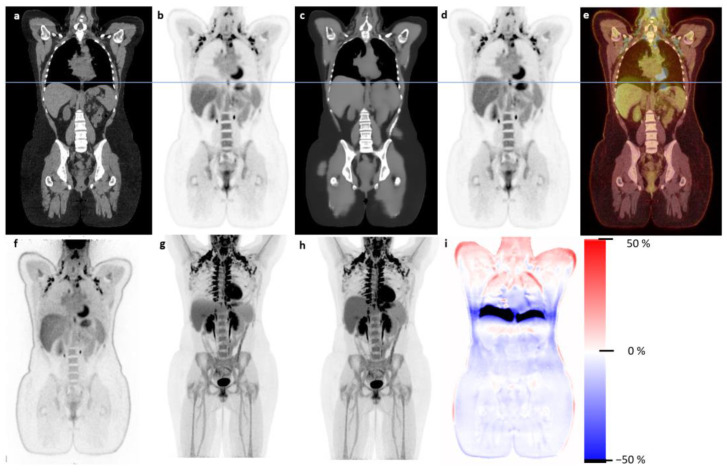
Illustrative sample patient with large banana artifact presented. Panels (**a**,**b**) show the normal CT (soft tissue window) and corresponding PET. The synthetic CT (sCT) and corresponding sPET are seen in (**c**,**d**). NAC PET is fused on top of the CT scan in (**e**), illustrating the mismatch between CT and emission data. The blue line represents the superior part of the liver at the time of CT scanning. Panel (**f**) shows the NAC PET used for synthesizing the sCT. (**g**,**h**) are MIPs of PET and sPET, respectively, and the voxel-wise relative difference between PET and sPET (**b**–**d**) is shown in (**i**). Note that the big deviation around the diaphragm is likely due to motion between CT scanning and PET emission.

**Figure 2 diagnostics-13-03661-f002:**
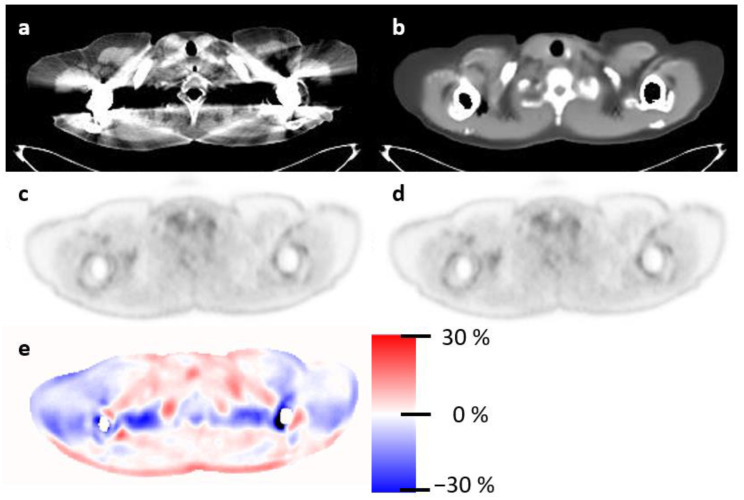
This patient has a double shoulder implant resulting in severe streaking artifact in he CT image (**a**). The synthetic CT (**b**) does not express these artifacts. This artifact propagates by attenuation correction to the PET data. Panel (**e**) illustrates this by showing the relative difference between PET (**c**) and sPET (**d**).

**Figure 3 diagnostics-13-03661-f003:**
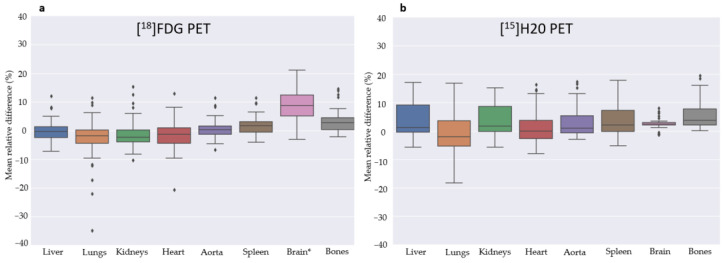
These Seaborn boxplots show the mean relative difference between PET and sPET for selected organs for [^18^F]FDG (**a**) and [^15^O]H_2_O (**b**) scans for the two respective test cohorts. * Note that the [^18^F]FDG training data did not include the brain in most of the scans. In contrast, the [^15^O]H_2_O training data from the LAFOV scanner always included the brain.

**Figure 4 diagnostics-13-03661-f004:**
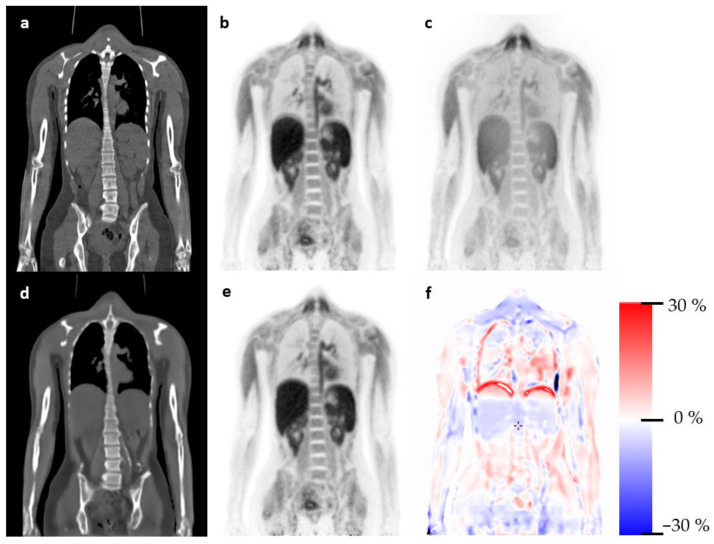
Representative patient scan using NAC PET from [^15^O]H_2_O for sCT synthesis. Panel (**a**,**b**) shows a sample coronal slice from the CT and PET images, respectively. Panel (**c**) presents the NAC PET used for synthesis of sCT (**d**). The PET image reconstructed using sCT for attenuation correction is shown in (**e**), with the relative differences in PET (**b**) shown in panel (**f**). Note the motion artifact between CT and NAC PET resulting in a shadow above the liver in (**b**), also seen in (**e**) as a change in reconstructed activity in the lower lungs due to breathing.

**Figure 5 diagnostics-13-03661-f005:**
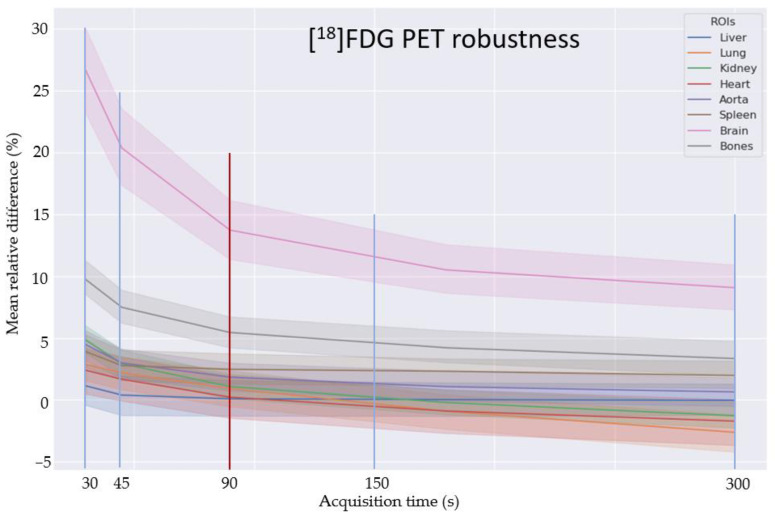
Robustness towards different acquisition times can be observed for most organs down to 45 s and stay below 5% deviation on average even for 30 s. The exception is the brain, which is not well represented in the training data. The deviation in bone is also higher, most likely due to the big impact on bone values due to motion, e.g., in the rib cage. The training cohort noise level corresponded to the 90 s test data in image quality (red line).

**Figure 6 diagnostics-13-03661-f006:**
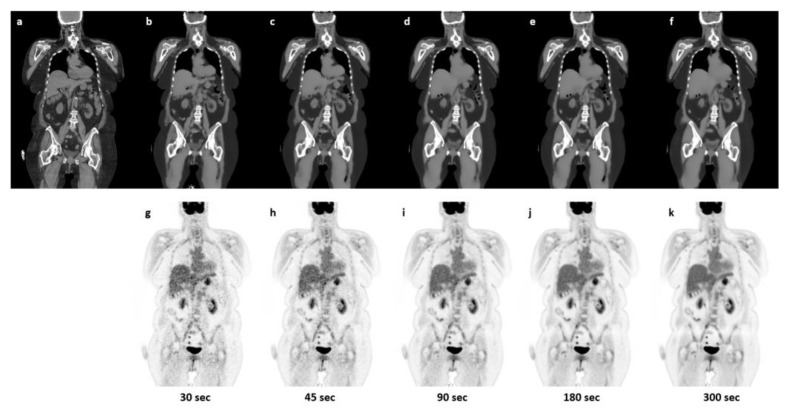
Sample patient illustration CT (**a**), sCT derived from [^18^F]FDG PET NAC for 30 s, 45 s, 90 s, 180 s, and 300 s (**b**–**f**) and reconstructed sPET data using the derived sCT with increasing frame sizes: 30 s, 45 s, 90 s, 180 s, and 300 s (**g**–**k**).

**Table 1 diagnostics-13-03661-t001:** Five cohorts are presented for Siemens Vision 600 PET/CT and Siemens Vision Quadra PET/CT with a tracer split of [^18^F]FDG and [^15^O]H_2_O.

Cohort	Radiotracer	PET/CT Scanner	Inclusion Period
Train (*n* = 858)	[^18^F]FDG	Siemens Vision 600	January 2021 to May 2022
Test (*n* = 78)	[^18^F]FDG	Siemens Vision 600	May 2022 to September 2022
Test (*n* = 36)	[^18^F]FDG	LAFOV Siemens Vision Quadra	November 2021 to August 2022
Train (*n* = 34)	[^15^O]H_2_O	LAFOV Siemens Vision Quadra	November 2021 to Marts 2023
Test (*n* = 12)	[^15^O]H_2_O	LAFOV Siemens Vision Quadra	October 2022 to June 2023

**Table 2 diagnostics-13-03661-t002:** Artifact score: 0 = none, 1 = minor (no clinical impact), 2 = medium, 3 = major (potential clinical impact). Overall image quality score: 0 = same image quality, 1 = insignificant difference, 2 = large difference with potential clinical impact.

Patient No.	Artifact	Overall Image Quality Score:	Quality Notes
2	Metal implant in tooth (1)	0	No significant impact
5		1	Possible metal artifact. Two Lesions on left flank easier seen on sPET
7	Tiny banana artifact on both recons (1)	0	
10	Banana artifact on PET (2)	1	sPET best
16	Arm movement, both recons score (1)	0	No clinical impact
19	Banana artifact on PET (2)	1	PET best
20	Tiny banana artifact on PET (1)	0	
22	Banana artifact on both recons (PET (2) sPET (1))	1	PET best
25	Streaking across abdomen on sPET (2)	1	No significant impact. PET best (1)
32	Lacking detail and streak in PET (1)	1	No significant impact. Very obese patient. sPET best (1)

## Data Availability

Data supporting reported results can be obtained via contact with the corresponding author upon reasonable request and legal approval. The data are not publicly available due to no public data sharing aggreement.

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
