# Peer review of "Attenuation Correction of Long Axial Field-of-View Positron Emission Tomography Using Synthetic Computed Tomography Derived from the Emission Data: Application to Low-Count Studies and Multiple Tracers"

_diagnostics, 2023, doi:10.3390/diagnostics13243661_

Round 1
Reviewer 1 Report
Comments and Suggestions for Authors
Regarding PET-CT imaging, we have here a well-written and relevant article on an important issue: ALARA irradiation criteria. In addition, authors endeavour to reduce motion and metal artefacts on PET-CT acquisitions.
However, clinical use takes precedence over technological tools. The CT scan is an integral part of the PET-CT examination: it gives the precise anatomical location of the metabolic disturbance assessed by absence or accumulation of tracer. Looking at sCT built by deep learning-AI presented here, physicians may be disappointed by the loss of 'imperfections' on native CT scans! Indeed, sCT 30 is familiar to the physician whereas sCT 300 looks so smooth... Fortunately, sCT 30 is always available when acquisition for 300 s is performed. So clinical validation is thus eagerly awaited.
Minor comments :
In general, there are too few images, and no [15O]-H2O image in the main manuscript : moving images from supplementary data and adding some more ?
Along the text : please replace sec by s.
Introduction :
“These previous studies typically have datasets in the range 25-220 that might be adequate for training a CNN, but increased sample size should increase robustness towards variation in body shapes” : please explain.
2.4 Synthetic CT generation :
2e-4, 2e-3, 1e-4… : does this mean 2.10-4, 2.10-3, 10-4 ? If so, please write clearly.
2.6 Data Analysis :
SSIM : please define.
Comments on the Quality of English LanguageThere are no major mistakes or errors but rereading is certainly needed.
Author Response
We would like to thank the reviewer for his reviewing the manuscript and appreciate the comments.
Comments and Suggestions for Authors
Regarding PET-CT imaging, we have here a well-written and relevant article on an important issue: ALARA irradiation criteria. In addition, authors endeavour to reduce motion and metal artefacts on PET-CT acquisitions.
However, clinical use takes precedence over technological tools. The CT scan is an integral part of the PET-CT examination: it gives the precise anatomical location of the metabolic disturbance assessed by absence or accumulation of tracer. Looking at sCT built by deep learning-AI presented here, physicians may be disappointed by the loss of 'imperfections' on native CT scans! Indeed, sCT 30 is familiar to the physician whereas sCT 300 looks so smooth... Fortunately, sCT 30 is always available when acquisition for 300 s is performed. So clinical validation is thus eagerly awaited.
We fully agree and intent to start clinical validation as soon as possible.
Minor comments :
In general, there are too few images, and no [15O]-H2O image in the main manuscript : moving images from supplementary data and adding some more ?
We have moved supplementary figures 1 and 2 into the main manuscript, thanks for the suggestion.
Along the text : please replace sec by s.
Done.
Introduction :
“These previous studies typically have datasets in the range 25-220 that might be adequate for training a CNN, but increased sample size should increase robustness towards variation in body shapes” : please explain.
We explained further and the sentence now reads:
These previous studies typically have datasets in the range 25-220 that might be adequate for training a CNN, but increased sample size should increase robustness towards variation in body shapes, artifacts and rare anatomy, as new patients are more likely to be represented in the training cohort [15].
2.4 Synthetic CT generation :
2e-4, 2e-3, 1e-4… : does this mean 2.10-4, 2.10-3, 10-4 ? If so, please write clearly.
This has been edited accordingly.
2.6 Data Analysis :
SSIM : please define.
SSIM has been written out, thanks for pointing this out.
Comments on the Quality of English Language
There are no major mistakes or errors but rereading is certainly needed.
The manuscript has been edited by a native English reader.
Reviewer 2 Report
Comments and Suggestions for Authors
The manuscript is very well-written and interesting, addressing an important topic in nuclear medicine. Introduction, methodology, results, discussion/conclusion are well-explained and I have only few suggestion for the authors:
- Please check all the acronyms (for example in line 35 correct LOFOV to LAFOV).
- I suggest a moderate revision of English language of the text.
Comments on the Quality of English LanguageI suggest a moderate revision of English language of the text.
Author Response
We would like to thank the reviewer for reviewing the manuscript.
Comments and Suggestions for Authors
The manuscript is very well-written and interesting, addressing an important topic in nuclear medicine. Introduction, methodology, results, discussion/conclusion are well-explained and I have only few suggestion for the authors:
- Please check all the acronyms (for example in line 35 correct LOFOV to LAFOV).
The manuscript have been edited for typing errors.
Comments on the Quality of English Language
I suggest a moderate revision of English language of the text.
The manuscript has been edited by a native English reader to improve the language, thanks for this suggestion.